# Definition and surgical timing in cauda equina syndrome–An updated systematic review

**Mohammad A. Mustafa**[1,2☯], **George E. Richardson**[1,2☯], **Conor S. Gillespie**[1,2]*,
**Abdurrahman I. Islim**[1,2], **Martin Wilby**[2], **Simon Clark**[2], **Nisaharan Srikandarajah**[2]

**1** School of Medicine, University of Liverpool, Liverpool, United Kingdom, **2** Department of Neurosurgery, The Walton Centre NHS Foundation Trust, Liverpool, United Kingdom

☯ These authors contributed equally to this work.
* hlcgill2@liv.ac.uk

## Abstract

### Study design

Systematic review.

### Objectives

To conduct a systematic review identifying existing definitions of cauda equina syndrome (CES) and time to surgery in the literature for patients with CES.

### Methods

A systematic review was conducted in accordance with the PRISMA statement. Ovid Medline, Embase, CINAHL Plus, and trial registries were searched from October 1st, 2016, to 30th December 2022, and combined with articles identified from a previous systematic review by the same authors (studies published 1990–2016).

### Results

A total of 110 studies (52,008 patients) were included. Of these only 16 (14.5%) used established definitions in defining CES, including Fraser criteria (n = 6), British Association of Spine Surgeons (BASS) (n = 5), Gleave and MacFarlane (n = 2), and other (n = 3). Most reported symptoms were urinary dysfunction (n = 44, 40%%), altered sensation in the perianal region (n = 28, 25.5%) and bowel dysfunction (n = 20, 18.2%). Sixty-eight (61.8%) studies included details on time to surgery. There was an increase in percentage of studies defining CES published in the last 5 years compared to ones from 1990–2016 (58.6% vs 77.5.%, *P* = .045).

### Conclusions

Despite Fraser recommendations, substantial heterogeneity exists in reporting of CES definitions, and a start point for time to surgery, with most authors using self-defined criteria. A consensus is required to define CES and time to surgery, to allow consistency in reporting and study analysis.

**Data Availability Statement:** All relevant data are within the article and its Supporting Information files.

**Funding:** The author(s) received no specific funding for this work.

**Competing interests:** The authors have declared that no competing interests exist.

## Introduction

Cauda equina syndrome (CES) is a rare condition caused by compression of the lumbosacral neve roots below the conus medullaris, and is a clinical-radiological diagnosis most commonly made using Magnetic Resonance Imaging (MRI) [1, 2]. Common symptoms and signs of CES include lower back pain, bladder and bowel dysfunction, motor weakness of the lower limbs, and saddle anaesthesia [3]. CES can be categorised on the basis of urinary symptoms- with CES associated with urinary retention and overflow incontinence being termed as complete (CES-Retention), and without being termed as incomplete (CES-I) [4]. CES is a surgical emergency, requiring rapid identification to prevent a high risk of morbidity [5], with many studies indicating that earlier surgical decompression leads to improved functional and long-term outcomes [6–9]. Cauda equina syndrome has a high morbidity and legal burden which necessitates a lower threshold for MRI [10].

Previous reviews have highlighted a lack of consensus among the definitions of CES used. A review by Fraser et al. in 2009 demonstrated marked inconsistencies in the literature and identified seventeen different definitions of CES, following which the authors suggested a standardised approach to define CES, The Fraser Criteria [11]. Nearly a decade later, a systematic literature review identified heterogenous outcome measures reported after surgery for CES, and affirmed that many studies did not employ a standardised definition for CES [2].

Timing of surgery for CES is of critical importance. However, there is no universally agreed definition for timing. For studies examining surgical outcomes in CES, it is unclear where they define when the timing started from, such as development of specific clinical symptoms, hospital admission, time of trauma/injury, or confirmation of CES on radiological imaging. A systematic review is required to elucidate existing definitions of symptoms and time to surgery for CES in literature within the last 20 years, after the advent of MRI.

### Review question

In studies of cauda equina syndrome where patients underwent surgery, what were the symptoms and signs used to define CES, and what was the definition of timing that the authors used?

### Objectives

The objective of this systematic literature review is to report symptoms used to define CES in existing studies, in addition to the reported timing definitions and intervals used for patients undergoing surgery.

## Material and methods

A systematic review was reported in accordance with the Preferred Reporting Items for Systematic Reviews and Meta-analyses (PRISMA) statement [12] and A systematic review critical appraisal tool (AMSTAR-2) [13]. The study was registered with PROSPERO (registration number: CRD42021261603).

### Search strategy

A literature search, last updated 30th December 2022, covered the time period 1st October 2016-30th December 2022 in the following study databases and registries: Medline (Ovid),

Embase (Ovid), and CINAHL Plus (EBSCO). The search strategy utilised for Embase, Medline, and CINAHL Plus can be found in S1 File. We scanned reference lists of included articles to identify additional studies in the review. Papers were limited to English language due to the feasibility of translation. Trial registries were searched for any ongoing or completed trials in surgery for CES.

### Study screening and selection

Articles identified from the search were transferred to the online platform Rayyan, a repository to facilitate de-duplication and independent screening of potential records [14]. After removal of duplicates, titles and abstracts were screened against the population, intervention, comparison, outcome, and study design (PICOS) criteria, where outcome referred to study characteristic, defined in Table 1 (S1 File) by two independent, blinded reviewers (GER and CSG). Following this, full-texts were screened by two independent, blinded reviewers (MAM and GER), to confirm manuscripts eligible for inclusion. All manuscripts eligible for inclusion were cross-checked by the senior author (NS).

If any disagreements occurred, an attempt was made to resolve this, and if discussion failed to lead to consensus, a senior author was consulted for clarification (NS). We combined the final records in this search with the studies used in a previously published systematic review, investigating reporting of outcomes following CES surgery. The review identified all papers of CES patients undergoing surgery from 1990–2016 [2]. The reason for limiting the search time period from 1990-present is to ensure that studies included are in the post-MRI era and that surgical management of CES is in line with current clinical practice. The inclusion criteria are shown in Table 1 (S1 File). The breakdown of existing criteria/guidelines for defining CES is provided in Table 2 (S1 File).

### Data extraction and synthesis

Results combined from our search and the previous systematic review were incorporated in a Microsoft Excel Spreadsheet. Data extraction was conducted independently and in duplicate by two authors (MAM and CSG) using a standardised pre-piloted data collection proforma. Data extracted included baseline patient demographics, cohort size, whether papers defined CES or not, and if so, what symptoms/criteria were employed, time to surgery, and definition of when time to surgery started. The primary outcome measure from the study characteristic was the symptoms used to define CES, and the time point at which measurement of time to surgery for CES started. The secondary outcome measures were number of studies providing surgical timing details, and whether reporting of CES definitions and time to surgery has become more standardised over the last 5 years.

### Statistical analysis

Frequencies were summarized using descriptive statistics. Continuous data was subject to a Kolmogorov-Smirnov test of normality, with normally distributed data being presented as a mean and standard deviation (SD) and skewed data being presented as a median and an inter-quartile range (IQR). Categorical variables were compared using the Chi-Squared test. Statistical analysis was conducted using SPSS Version 26.0 with figures being generated in R V4.0.2

### Ethical approval

As this is a systematic review including published studies presenting data at a study level and narratively, individual participant consent was not required.

## Results

### Study selection process

The PRISMA flowchart in Fig 1 describes the study selection process. The total number of studies included following our search was 35, this combined with the results of the previous review resulted in a total of 96 studies being included. A re-search of all three databases was conducted which identified a further 14 articles which were included. S1 File provides the complete list of included studies.

### Study and patient characteristics

Ninety-five (82.6%) studies were retrospective, with 15 (13.4%) being prospective. Most of the studies were conducted in Europe, (n = 63, 54.8%), with 25 (21.7%) from North America, 21 (18.3%) from Asia and 1 (0.87%) from South America. The total number of patients included in this review was 52,008. The study characteristics, and definitions are outlined in Table 3 (S1 File).

### CES etiology

A total of 99 studies (90.0%) reported the underlying etiology for CES. Disc herniation was reported as the most common etiology (n = 67, 60.9%). The remaining studies reported trauma (n = 9, 8.2%), degenerative (n = 9, 8.2%), tumor (n = 6, 5.5%), postoperative complication (n = 3, 2.7%), and other (n = 5, 4.5%) as underlying etiology for CES.

### CES definition

CES was defined in 72 (65.5%) studies, with 16 (14.5%) of those using established guidelines. The most commonly used guidelines were Fraser (n = 6, 5.5%), followed by BASS guidelines (n = 5, 4.5%), Gibbons (n = 2, 1.8%), Gleave and MacFarlane (n = 2, 1.8%) and Shi (n = 1, 0.9%). The remaining 56 studies that defined CES did not use pre-existing definitions/classifications and used study specific criteria instead. CES was not defined in 43 studies.

Among the studies that did not use pre-existing definitions/classifications, a combination of symptoms and signs were used to define CES. The most common symptoms/signs used were urinary dysfunction (n = 53, 48.2%), altered sensation in the perianal region (n = 28, 25.5%), loss of rectal tone (n = 19, 17.3%), bowel dysfunction (n = 20, 18.2%), sensory disturbance in the lower limb (n = 16, 14.5%), motor dysfunction in the lower limb (n = 15, 13.6%), sexual dysfunction (n = 14, 12.7%), and sciatica (n = 7, 6.4%). The median (IQR) number of symptoms used to define CES was 3 (2–5).

There was a statistically significant increase in the percentage of studies defining CES over the last 5 years compared to the ones published in 1990–2017, with 58.5% of studies defining CES between the years 1990–2017 (n = 41), and 77.5% (n = 31) defining CES between the years 2018–2022, *P = .045.*

There was a statistically significant increase in percentage of studies using pre-existing criteria over the time period 2018–2022 compared to the ones published in 1990–2017, with 7.3% of studies using criteria between 1990–2017 (n = 3), and 40.6% (n = 13) of studies using them between the years 2017–2022, *P = < .001.* Fig 2 illustrates the reporting of CES symptoms over time.

All the studies employing the use of the Gleave and MacFarlane, Fraser and British Association of Spine Surgeons (BASS) guidelines were published within the last five years, while no studies published within the last five years used the Gibbons or Shi classifications. Studies not using pre-existing guidelines published from 1990–2016 used a median of 3 (IQR: 1–5)

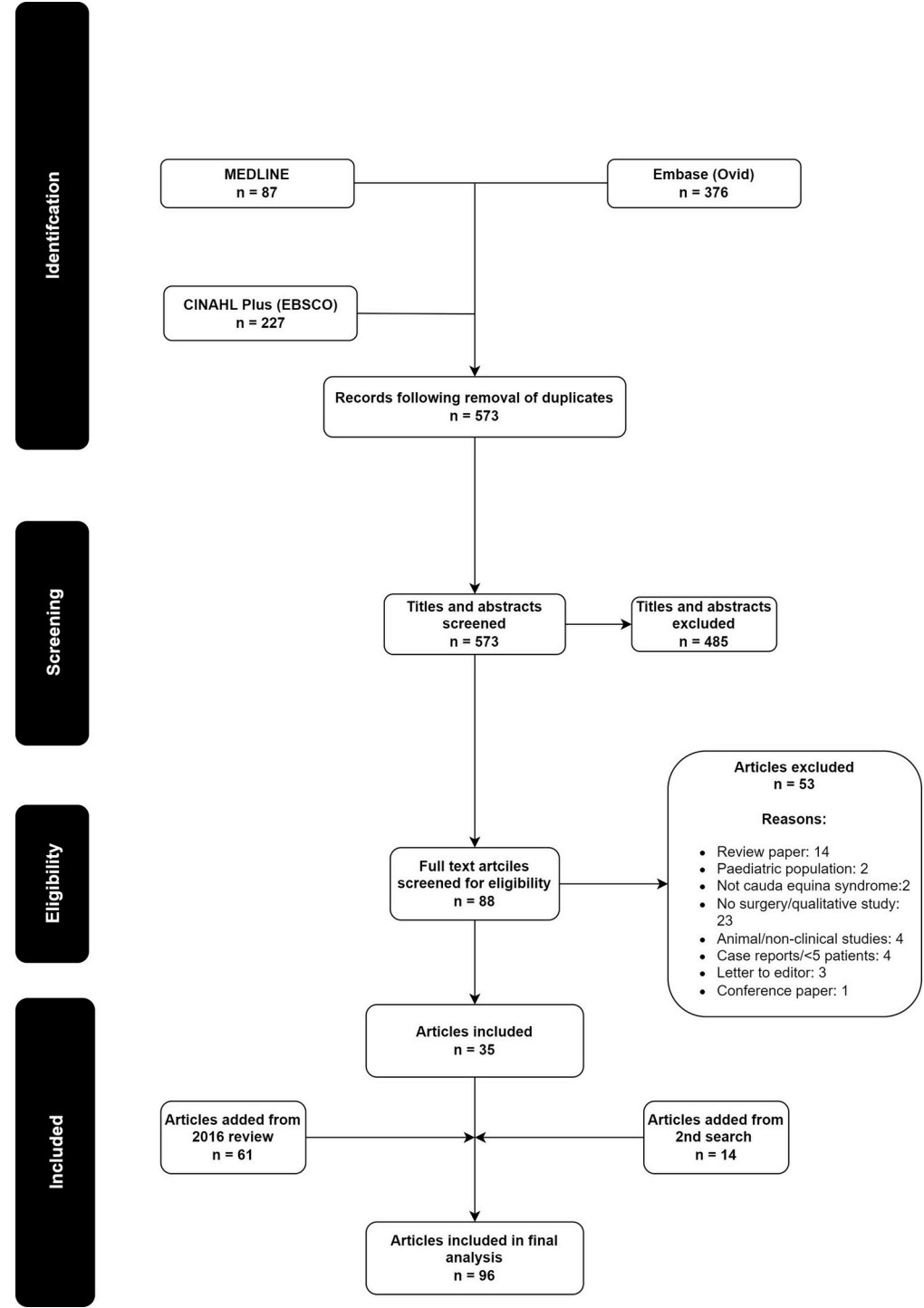

**Fig 1. PRISMA flowchart detailing the search, screening process, and final inclusion of studies.**

symptoms to define CES, however studies published between 2016–2021 used a mean of 4 (IQR: 2.5–5.5) symptoms to define CES.

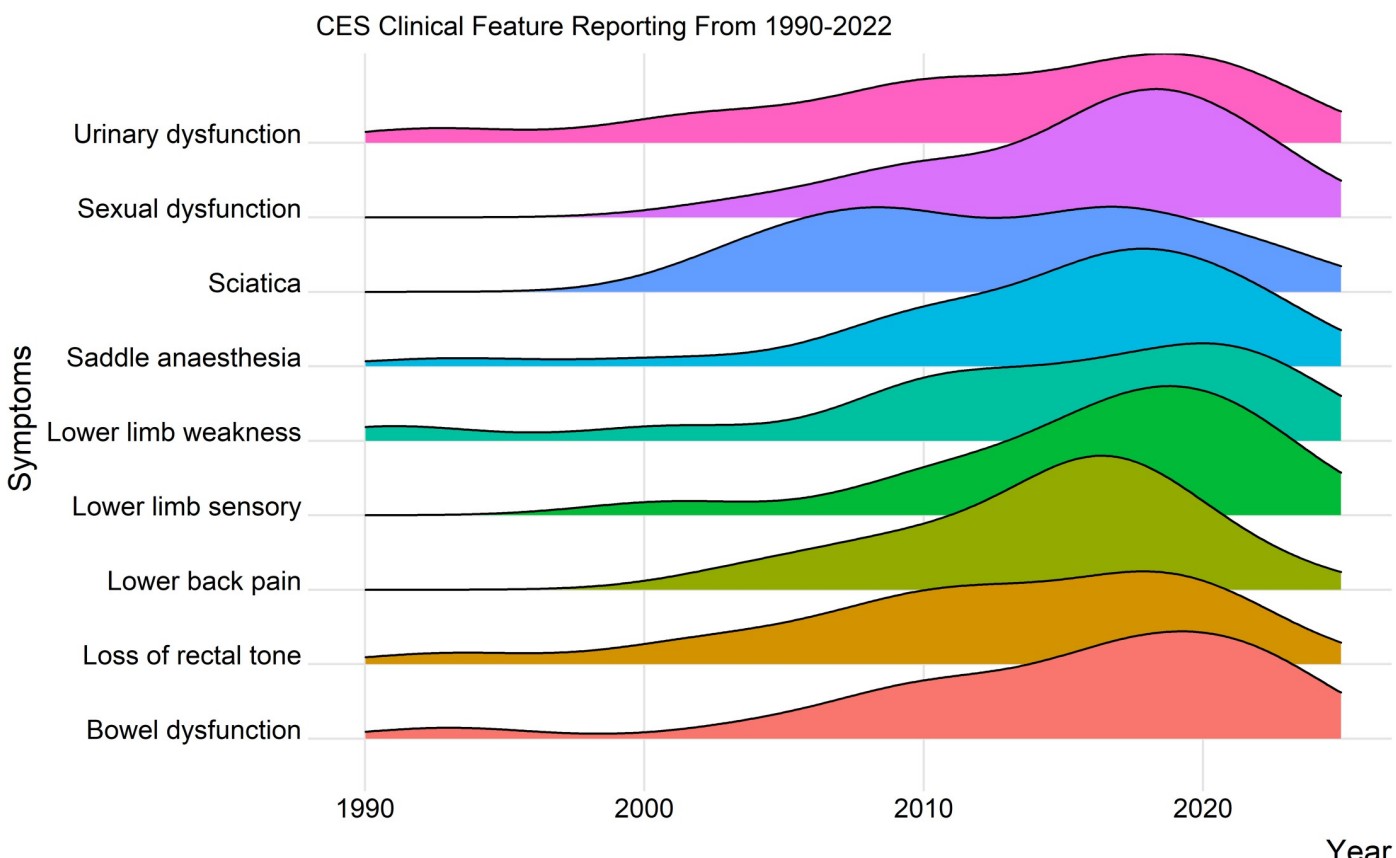

**Fig 2. Ridge-line plot representing reporting of symptoms and guidelines to define CES, over time.**

## Time to surgery details

A total of 68 (61.8%) studies included details of time to surgery, with 62 (43.6%) stating what they considered a starting point to measure the time to surgery. The remaining five studies provided details for time to surgery but did not mention a start point from which they measured this time. A study-specific breakdown of time to surgery details is provided in S1 File. Forty (36.4%) studies defined timing from symptom onset, which included urinary symptoms (n = 16), bowel dysfunction (n = 10), lower limb weakness/sensory disturbance (n = 6), saddle anaesthesia (n = 7), back pain (n = 6), and sexual dysfunction (n = 5). Other time points included hospital admission (n = 7), presentation to first doctor (n = 4), time of injury (used in traumatic CES) (n = 4), radiological confirmation/surgeon's decision to operate (n = 5), and time of referral (n = 2). Of the studies published between 1990–2017, 42 (60%%) included time to surgery details and for studies published between 2018 onwards, 20 (50%%) included time to surgery details.

## Surgical method

A total of 75 (68.2%) studies reported the surgical method used. Laminectomy was the most common method (n = 24, 21.8%), with other methods reported including microdiscectomy (n = 16, 14.5%), laminectomy and discectomy (n = 14, 12.7%), laminectomy and instrumentation (n = 12, 10.9%), and other (n = 6, 5.5%).

All papers reporting surgical method for trauma (n = 8) employed laminectomy and instrumentation as the main surgical method. Microdiscectomy (n = 16) was exclusively reported in studies with a disc herniation etiology. Other surgical methods included untethering surgery, tumour resection, and lumboperitoneal shunt insertion.

## Discussion

This systematic review demonstrates that the heterogeneity of CES symptom definition continues to persist between 1990–2022, with only 3 in 5 studies stating how they defined CES. The reporting of time to surgery is poor with only 1 in 2 studies stating when they start time to surgery from, with 39% of published studies not including any timing details at all.

As highlighted in an earlier review by Fraser et al, studies are not useful to clinicians in making a diagnosis and managing CES unless they offer information about its presentation and cause. Due to the wide variety of presentations in CES patients, it is very challenging to find an all-encompassing definition. There has been a marked increase in studies defining CES over the last five years, however there was no improvement in the use of pre-existing guidelines over this time period. Authors have indicated differences in the importance of specific symptoms, such as Gleave and Macfarlane et al. classifying urinary symptoms as the reference to classify severity of CES, while Tandon and Sankaran [15, 16] classify CES based on back problems, bladder dysfunction, and sciatica. Although the literature consists of defined criteria, such as the Fraser criteria, Gleave and Macfarlane, Gibbons, and BASS guidelines, their adoption was limited, with only 16 out of 110 studies using pre-existing guidelines and definitions. All studies that used the Fraser criteria were published in the last six years, indicating a slow but promising uptake of these definitions. The disparity in using varying definitions might stem from differences in perceived importance of these symptoms by clinicians, coupled with difficulty in assessing them uniformly due to the symptoms being either patient-reported, clinician assessed or both.

A core outcome set developed by Srikandarajah et al developed a list of 16 outcomes that are essential to be reported in CES studies, following an international consensus process consisting of patients and health care professionals [17]. Due to the varying nature in presentation of CES, a consensus process might be the way forward in outlining a set of symptoms/signs, which define CES specifically, in addition to the reporting of the start point for time to surgery [18].

There are numerous studies that examine the effect of timing for surgical decompression in CES and its association with improved outcomes [6, 8, 9, 19], however no studies to date have investigated the definition of when timing begins, in addition to whether studies report patients that received surgery within a specific time interval, or an overall mean or median statistic. Early decompression has been demonstrated to improve bladder function, overall function, and quality of life, highlighting the importance of timing intervals in CES, which is considered a neurosurgical emergency [20, 21]. As time to surgery is a crucial factor in CES management, differing definitions of it make comparison of results between studies challenging, precluding pooled analysis and meta-analysis. Among the studies that do define a specific time point, most of them use development of symptoms as the time point. Due to the heterogenous nature of symptoms reported to define CES, it is often not possible to elucidate what impact these timing intervals have.

Other studies that did not define timing from symptoms, used equally heterogenous definitions such as hospital admission, presentation to first doctor and radiological confirmation. These checkpoints vary considerably depending on characteristics of the healthcare system and the underlying aetiology of CES [22]. A major limitation among reporting time to surgery arises from the design of these studies being included. As most included studies were

retrospective, it is more challenging to record this information at presentation. Although these studies aim to demonstrate the effect of time to surgery on patient outcomes, lack of uniformity in defining these time-points make them less informative to clinicians worldwide.

The underling etiology of CES adds to the heterogeneity in defining time to surgery. Nearly 60% of included studies predominantly reported disc herniation as the cause of CES with trauma, tumour and degenerative pathologies being reported in <10% of studies. The timing of presentation to hospital differs vastly for the above etiologies, precluding a uniform, 'one-size fits all' definition for time to surgery.

Another aspect of CES surgery which is less commonly explored is the time of surgery. Only two studies in the literature report on outcomes relating to the time of surgery [21, 22]. Both are single centre retrospective studies conducted in the United Kingdom and report higher rates of intra-operative complications for decompression surgeries carried out overnight. Additionally, Mirza et al reported no significant difference in individual patient outcome at 6 months [21]. With the longstanding philosophy of "time is spine", time of surgery needs to be a factor in scheduling operations, especially if surgeries out of hours provide no added benefit to the ultimate outcome at patient level.

Due to a lack of consistency among defining time to surgery in CES, there is a need for researchers in future work to consider appropriate and uniform time definitions, and to specify when timing was started. The approach to define a specific time point, considering its perceived importance and ease of measurement, should be done using a consensus process that considers ideas and viewpoints from key stakeholders.

## Limitations

Our study has three limitations. First, we included articles only included in English. Secondly, due to heterogeneity of the data reported, we did not carry out a meta-analysis, and finally, we did not conduct risk of bias or quality assessments as we conducted a descriptive analysis.

In conclusion, significant heterogeneity still exists in reporting of both symptoms used to define CES and start point for timing to surgery. There is a clear need to develop a consensus for a CES definition according to symptoms/signs. This will allow comparison of interventions, facilitate meta-analysis, and define a critically important spinal emergency condition with long lasting implications.

## Supporting information

**S1 Checklist. PRISMA 2020 checklist.**
(DOCX)

**S1 File.**
(DOCX)

**S1 Data.**
(XLSX)

## Author Contributions

**Conceptualization:** Conor S. Gillespie, Abdurrahman I. Islim, Martin Wilby, Simon Clark, Nisaharan Srikandarajah.

**Data curation:** Mohammad A. Mustafa, George E. Richardson, Conor S. Gillespie.

**Formal analysis:** Mohammad A. Mustafa, George E. Richardson, Conor S. Gillespie.

**Funding acquisition:** Mohammad A. Mustafa, George E. Richardson, Conor S. Gillespie, Nisaharan Srikandarajah.

**Investigation:** Mohammad A. Mustafa, George E. Richardson, Conor S. Gillespie.

**Methodology:** Mohammad A. Mustafa, George E. Richardson, Conor S. Gillespie, Nisaharan Srikandarajah.

**Project administration:** Mohammad A. Mustafa, George E. Richardson, Conor S. Gillespie, Abdurrahman I. Islim, Nisaharan Srikandarajah.

**Resources:** Mohammad A. Mustafa, George E. Richardson, Conor S. Gillespie, Nisaharan Srikandarajah.

**Software:** Mohammad A. Mustafa, George E. Richardson, Conor S. Gillespie, Nisaharan Srikandarajah.

**Supervision:** Mohammad A. Mustafa, Conor S. Gillespie, Abdurrahman I. Islim, Martin Wilby, Simon Clark, Nisaharan Srikandarajah.

**Validation:** Mohammad A. Mustafa, Conor S. Gillespie, Abdurrahman I. Islim, Martin Wilby, Simon Clark, Nisaharan Srikandarajah.

**Visualization:** Mohammad A. Mustafa, George E. Richardson, Conor S. Gillespie, Nisaharan Srikandarajah.

**Writing – original draft:** Mohammad A. Mustafa, George E. Richardson, Conor S. Gillespie, Abdurrahman I. Islim, Martin Wilby, Simon Clark, Nisaharan Srikandarajah.

**Writing – review & editing:** Mohammad A. Mustafa, George E. Richardson, Conor S. Gillespie, Abdurrahman I. Islim, Martin Wilby, Simon Clark, Nisaharan Srikandarajah.

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
