## [Decision Letter · Decision Letter 0]

6 Jul 2022

PONE-D-22-00142Definition and surgical timing in CES - A systematic reviewPLOS ONE

Dear Dr. Gillespie,

Thank you for submitting your manuscript to PLOS ONE. After careful consideration, we feel that it has merit but does not fully meet PLOS ONE’s publication criteria as it currently stands. Therefore, we invite you to submit a revised version of the manuscript that addresses the points raised during the review process.

We look forward to receiving your revised manuscript.

Kind regards,

Alfio Spina, M.D.

Academic Editor

PLOS ONE

Journal Requirements:

https://journals.plos.org/plosone/s/file?id=ba62/PLOSOne_formatting_sample_title_authors_affiliations.pdf".

“The authors did not receive any external funding for the completion of this study. The manuscript production fees were covered by a grant from the University of Liverpool library. CSG is supported by a grant from the Wolfson Foundation”

Reviewers' comments:

Reviewer's Responses to Questions

**Comments to the Author**

1. Is the manuscript technically sound, and do the data support the conclusions?

Reviewer #1: Yes

Reviewer #2: Partly

2. Has the statistical analysis been performed appropriately and rigorously? 

Reviewer #1: Yes

Reviewer #2: Yes

3. Have the authors made all data underlying the findings in their manuscript fully available?

Reviewer #1: Yes

Reviewer #2: Yes

4. Is the manuscript presented in an intelligible fashion and written in standard English?

Reviewer #1: Yes

Reviewer #2: Yes

5. Review Comments to the Author

Reviewer #1: This is an excellent review of a timely topic. If the same language and definition of time is not used then it is impossible to determine the results across studies. This work will highlight that need.

Reviewer #2: This is a very interesting study concerning the definition and surgical timing in cauda equinal syndrome. But the varieties of the syndrome were highly correlated to the etiologies of disease. The surgical timing also depended on the diseases themselves.

The authors should add the parameters of etiologies as a crucial factor for the analysis.

6. PLOS authors have the option to publish the peer review history of their article (what does this mean?). If published, this will include your full peer review and any attached files.

Reviewer #1: No

Reviewer #2: No

---

## [Author Response · Author response to Decision Letter 0]

18 Aug 2022

Dr. Alfio Spina 

Academic Editor, 

PLOS One 

Dear Alfio Spina,

Re: Definition and surgical timing in Cauda Equina Syndrome – A systematic review 

Thank you considering our revised manuscript entitled “Definition and surgical timing in Cauda Equina Syndrome – A systematic review” for publication in PLOS One. We thank the reviewers for their constructive comments and feedback. We hope we have addressed their concerns adequately in our revision of the manuscript and attach our point-by-point responses below. We believe that this manuscript is an important addition to the literature and the revisions have strongly enhanced this message. 

I would like to confirm again that with this submission, all study authors declare that they have reviewed the final manuscript, approved the contents of it and that the requirements for authorship have been met by all named authors. We can guarantee that the work is not being considered for publication by any other journal and that no previous work we’ve published (apart from meeting abstracts) overlap with the current work. We have not received any external funding for the completion of this work and have no relevant conflicts of interest to report.

I thank you once again for considering our manuscript for publication and look forward to hearing from you soon. 

Yours Sincerely,

Conor S. Gillespie 

MPhil student 

Institute of Systems, Molecular and Integrative Biology (ISMIB) 

University of Liverpool, UK 

Email: hlcgill2@liv.ac.uk

Response to Associate Editor comments:

1. Please ensure that your manuscript meets PLOS ONE’s style requirements, including those for file naming. 

A: Thank you for highlighting this discrepancy and providing the link to the author and affiliation style format. We have now edited the title page in line with PLOS ONE’s style requirements. 

2. Please provide additional details regarding participant consent. In the ethics statement in the Methods and online submission information, please ensure that you have specified (1) whether consent as informed and (2) what type you obtained (for instance, written or verbal, how it was documented and witnessed). If your study included minors, state whether you obtained consent from parents or guardians. If the need for consent was waived by the ethics committee, please include this information. 

A: Thank you for highlighting this point. As this is a systematic review including published studies and presenting data at a study level, individual participant consent was not required. We have specified the same in the ethics statement in the Methods (Lines 156-159, Revised Manuscript with Track Changes) and the online submission information field. 

3. Thank you for stating the following in the Funding Section of your manuscript: “The authors did not…grant from the Wolfson Foundation”. 

We note that you have provided funding information that is not currently declared in your Funding Statement. However, funding information should not appear in the Acknowledgements section or other areas of your manuscript. We will only publish funding information present in the Funding Statement section of the online submission form. Please remove any funding-related text from the manuscript and let us know how you would like to update your Funding Statement. Currently, your Funding Statement reads as follows: “The author(s) received no specific funding for this work”. 

A: Thank you for highlighting this discrepancy between the Funding Section of our manuscript and the Funding Statement. For consistency we have removed the funding section from our manuscript. We would like the Funding Statement to read as “The author(s) received no specific funding to complete this work. The manuscript publication costs are funded by the University of Liverpool”.

4. We note that you have indicated that data from this study are available upon request. PLOS only allows data to ab. We note that you have indicated that data from this study are available upon request. PLOS only allows data to be available upon request if there are legal or ethical restrictions on sharing data publicly. For more information on unacceptable data access restrictions, please see http://journals.plos.org/plosone/s/data-availability#loc-unacceptable-data-access-restrictions.

A: a) We confirm that there are no ethical or legal restrictions on sharing our dataset.

(b) Alongside the revised manuscript, we will upload a minimal anonymized data set containing data points used in our analysis. The dataset will be uploaded as a Microsoft Excel file.

5. Review Comments to the Author

Reviewer #1: This is an excellent review of a timely topic. If the same language and definition of time is not used then it is impossible to determine the results across studies. This work will highlight that need.

Reviewer #2: This is a very interesting study concerning the definition and surgical timing in cauda equinal syndrome. But the varieties of the syndrome were highly correlated to the etiologies of disease. The surgical timing also depended on the diseases themselves.

The authors should add the parameters of etiologies as a crucial factor for the analysis.

A: Reviewer #1 – Thank you very much for reviewing our paper and the positive comment. We wholeheartedly agree with this and recognise the need for establishing a consensus for CES being a relevant topic. 

Reviewer #2 – Thank you very much for reviewing our paper and highlighting the importance of CES etiologies in our analysis. We have now included a section with descriptive statistics outlining the etiologies in included papers in our Results section (Line 174-178, Revised Manuscript with Track Changes).

---

## [Decision Letter · Decision Letter 1]

19 Dec 2022

PONE-D-22-00142R1Definition and surgical timing in Cauda Equina Syndrome – A systematic reviewPLOS ONE

Dear Dr. Gillespie,

Thank you for submitting your manuscript to PLOS ONE. After careful consideration, we feel that it has merit but does not fully meet PLOS ONE’s publication criteria as it currently stands. Therefore, we invite you to submit a revised version of the manuscript that addresses the points raised during the review process.

We look forward to receiving your revised manuscript.

Kind regards,

Andreas K Demetriades, MBBChir, MPhil, FRCSEd, FEBNS.

Academic Editor

PLOS ONE

Additional Editor Comments (if provided):

Thanks for your patience.

The handling editor had to change due to previous delays, and reviewers not responding, necessitating a fresh restart of reviews.

There are still significant shortcomings, as per peer review and we hope you find the comments constructive.

Reviewers' comments:

Reviewer's Responses to Questions

**Comments to the Author**

1. If the authors have adequately addressed your comments raised in a previous round of review and you feel that this manuscript is now acceptable for publication, you may indicate that here to bypass the “Comments to the Author” section, enter your conflict of interest statement in the “Confidential to Editor” section, and submit your "Accept" recommendation.

Reviewer #1: All comments have been addressed

Reviewer #2: All comments have been addressed

Reviewer #3: (No Response)

Reviewer #4: All comments have been addressed

2. Is the manuscript technically sound, and do the data support the conclusions?

Reviewer #1: Yes

Reviewer #2: Yes

Reviewer #3: Yes

Reviewer #4: Partly

3. Has the statistical analysis been performed appropriately and rigorously? 

Reviewer #1: Yes

Reviewer #2: Yes

Reviewer #3: Yes

Reviewer #4: No

4. Have the authors made all data underlying the findings in their manuscript fully available?

Reviewer #1: Yes

Reviewer #2: (No Response)

Reviewer #3: Yes

Reviewer #4: Yes

5. Is the manuscript presented in an intelligible fashion and written in standard English?

Reviewer #1: Yes

Reviewer #2: Yes

Reviewer #3: Yes

Reviewer #4: Yes

6. Review Comments to the Author

Reviewer #1: The authors present an acceptable version of their paper. The authors present an acceptable version of their paper. The

Reviewer #2: The authors had already addressed all the response to comments recommended by the reviewer. The manuscript should be published.

Reviewer #3: Thank you for giving me the opportunity to review the present manuscript. The authors performed a systematic review of the literature on the definitions and timing of intervention for cauda equina syndrome. The manuscript has scientific merit as it illustrates the lack of adoption of established knowledge and "guidelines" on the topic, probably due to the lack of a consensus on the subject.

Please find my comments below:

1) Title: I suggest that the title be revised to illustrate that this is an "updated" systematic review

2) Abstract:

a) Study design: narrative review -> please revise

b) Be consistent in your use of "n" for study number and the use of percentages

3) Methods:

a) Search strategy: The literature search is outdated (last update: 30th April 2021). Please re-run the literature search to identify articles published in the last 1.5 years.

b) Search strategy: Although the PRISMA guidelines only require the strategy of at least 1 database to be presented, I suggest that the search strategy for all databases is presented in Table S1.

c) Study screening and selection: Lines 123-126, please revise the sentence.

d) Statistical analysis: please mention the statistical tests that yielded the p values mentioned later in the text

e) Quality assessment: please provide a quality assessment paragraph as mentioned in your PRISMA protocol. A paragraph with the results should also be provided in the "Results" section.

4) Results:

a) I believe that combining some of the data of the supplementary tables to create 1-2 Tables presented in the main text with the basic characteristics of the included studies will help readers better understand the findings. (suggestion)

b) Be consistent in your use of "n" for study number and the use of percentages

c) Please provide statistical significance values when comparing the two periods (1990-2016 & 2016-2021). For example here: "There was an increase in percentage of studies using pre-existing criteria over the time period

205 2016-2021 compared to the ones published in 1990-2016, with 9.4% of studies using criteria

206 between 1990-2016 (n=3), and 38.7% (n=12) of studies using them between the years 2016-

207 2021."

5) Discussion:

a) Make sure to include references consistently when mentioning other studies.

Reviewer #4: Authors present a revised version of the narrative review of definiton and surgical timing in cauda equina syndrome. Methodology of study selection is fairly well described. Authors conclude that there is currently no concensus , either on definiton nor on how early is the early timing in management of cauda equina syndrome.

One major drawback of this study is that it does not discuss one very important aspect of the studies which were analyzed - the etiology. Cauda equina syndrome can appear due to tumor compression - primary or metastasis, due to trauma, infection or degenerative disease. We are sure that there would be certain differences in the management, timing and surgical treatment in general; what authors did in the revision is just information of which etiologies in which percentage led to the syndrome. Furthermore, this review provides a mere description of the evaluated studies, without any comparison between the studies; surgical philosophy is not discussed at all (decompression surgery, decompression + stabilization), and one very important clinical aspect - not only time of the surgery in terms of from onset of symptoms to surgery, but also time of the day of the surgery - were there any studies which analyze this very important aspect and its correlation to timing? - for example:

Baig Mirza A, Velicu MA, Lyon R, Vastani A, Boardman T, Al Banna Q, Murphy C, Kellett C, Vasan AK, Grahovac G. Is Cauda Equina Surgery Safe Out-of-Hours? A Single United Kingdom Institute Experience. World Neurosurg. 2022 Mar;159:e208-e220. doi: 10.1016/j.wneu.2021.12.028. Epub 2021 Dec 14. PMID: 34915208.

Demetriades AK. Cauda equina syndrome - from timely treatment to the timing of out-of-hours surgery. Acta Neurochir (Wien). 2022 May;164(5):1201-1202. doi: 10.1007/s00701-022-05174-1. Epub 2022 Mar 30. PMID: 35352153.

Are there any information of complication rates of these surgeries? To simply describe timing without description of outcome does not make a lot of sense, so I suggest to include outcome assesment following surgery in correlation to time, and to include into Discussion and comment on current literature:

Woodfield J, Lammy S, Jamjoom AA, Fadelalla MA, Copley PC, Arora M, Glasmacher SA, Abdelsadg M, Scicluna G, Poon MT, Pronin S, Leung AH, Darwish S, Demetriades AK, Brown J, Eames N, Statham PF, Hoeritzauer I; UCES Study Collaborators; British Neurosurgical Trainee Research Collaborative. Demographics of Cauda Equina Syndrome: A Population Based Incidence Study. Neuroepidemiology. 2022 Oct 31. doi: 10.1159/000527727. Epub ahead of print. PMID: 36315989.

I urge authors to analyze the following manuscript and compare their results to this review:

Epstein NE. Review/Perspective: Operations for Cauda Equina Syndromes - "The Sooner the Better". Surg Neurol Int. 2022 Mar 25;13:100. doi: 10.25259/SNI_170_2022. PMID: 35399881; PMCID: PMC8986648.

7. PLOS authors have the option to publish the peer review history of their article (what does this mean?). If published, this will include your full peer review and any attached files.

Reviewer #1: No

Reviewer #2: No

Reviewer #3: No

Reviewer #4: No

---

## [Author Response · Author response to Decision Letter 1]

1 Feb 2023

Response to review comments to the author:

Reviewer #1: The authors present an acceptable version of their paper. The authors present an acceptable version of their paper. 

Reviewer #2: The authors had already addressed all the response to comments recommended by the reviewer. The manuscript should be published.

A: Reviewer #1 – Thank you very much for reviewing our paper and the positive comment. We wholeheartedly agree with this and recognise the need for establishing a consensus for CES being a relevant topic. 

Reviewer #3: 

1. Title: I suggest that the title be revised to illustrate that this is an “updated” systematic review

A: Thank you for highlighting this point. We have now added “updated” to our title to reflect our choice to build on the systematic review published by Mr. Nisaharan Srikandarajah, titled, “Outcomes Reported After Surgery for Cauda Equina Syndrome” (2018, Spine), as it followed the same inclusion criteria to our review. 

2. Abstract:

a) Study design: narrative review -> please revise

A: Thank you for highlighting this. We have now changed the Study Design section in the Abstract to “Systematic review”. 

b) Be consistent in your use of “n” for study number and the use of percentages 

A: For consistency we have now used the total study number as “n” throughout this section and changed the percentages to reflect the same.

3. Methods:

a) Search strategy: The literature search is outdated (last update: 30th April 2021). Please re-run the literature search to identify articles published in the last 1.5 years.

A: We have now updated the systematic review and included all articles identified as per our last search (30th December 2022). We have now added a further 14 articles to our results.

b) Search strategy: Although the PRISMA guidelines only require the strategy of at least 1 database to be presented, I suggest that the search strategy for all databases is presented in Table S1.

A: Thank you for the above suggestion. We have now added the search strategy for all three databases to the electronic supplementary material.

c) Study screening and selection: Lines 123-126, please revise the sentence.

A: Thank you for highlighting the above. We have now re-structured lines 123-126, Methods, for clarity. The section now reads as follows: “After removal of duplicates, titles and abstracts were screened against the population, intervention, comparison, outcome, and study design (PICOS) criteria, where outcome referred to study characteristic, defined in Table 1 by two independent, blinded reviewers (GER and CSG). Following this, full-texts were screened by two independent, blinded reviewers (MAM and GER), to confirm manuscripts eligible for inclusion. All manuscripts eligible for inclusion were cross-checked by the senior author (NS).”

d) Statistical analysis: please mention the statistical tests that yielded the p values mentioned later in the text

A: Thank you for highlighting the lack of an appropriate statistical mentioned. We have now stated that categorical variables were compared using the Chi-Squared test (Line 153, Results, Statistical Analysis).

e) Quality assessment: please provide a quality assessment paragraph as mentioned in your PRISMA protocol. A paragraph with the results should also be provided in the "Results" section.

A: Thank you for highlighting the above. We decided against conducting risk of bias and quality assessments for this review as we did not conduct any synthetic analysis and only used descriptive methods to describe the heterogeneity in CES definition and time to surgery details. We included all studies that matched our inclusion criteria due to the heterogenous literature consisting of predominantly retrospective studies. To reflect the above we have now submitted an amendment to our PROSPERO form. 

4. Results:

a) I believe that combining some of the data of the supplementary tables to create 1-2 Tables presented in the main text with the basic characteristics of the included studies will help readers better understand the findings. (suggestion)

A: Thank you for the above suggestion. We have now combined some of the data in the supplementary tables and incorporated Table S4 (previously S2), into the main text of the manuscript. We hope this adds to the manuscript by providing more information about the included papers to the readers. 

b) Be consistent in your use of "n" for study number and the use of percentages

A: Thank you pointing out the above. We have used the total number of studies as the “n” throughout the results section for clarity. 

c) Please provide statistical significance values when comparing the two periods (1990-2016 & 2016-2021). For example here: "There was an increase in percentage of studies using pre-existing criteria over the time period

205 2016-2021 compared to the ones published in 1990-2016, with 9.4% of studies using criteria 206 between 1990-2016 (n=3), and 38.7% (n=12) of studies using them between the years 2016-207 2021."

A: We have now added p-values from the Chi-Squared test conducted to compare the categorical variables. We hope this emphasises the difference in the above rates of studies defining CES and reporting time to surgery. 

5. Discussion: 

a) Make sure to include references consistently when mentioning other studies

A: Thank you for highlighting the above. We have revised the discussion section as per comments from all reviewers and included consistent references throughout. 

Reviewer #4: Authors present a revised version of the narrative review of definiton and surgical timing in cauda equina syndrome. Methodology of study selection is fairly well described. Authors conclude that there is currently no concensus , either on definiton nor on how early is the early timing in management of cauda equina syndrome.

One major drawback of this study is that it does not discuss one very important aspect of the studies which were analyzed - the etiology. Cauda equina syndrome can appear due to tumor compression - primary or metastasis, due to trauma, infection or degenerative disease. We are sure that there would be certain differences in the management, timing and surgical treatment in general; what authors did in the revision is just information of which etiologies in which percentage led to the syndrome. Furthermore, this review provides a mere description of the evaluated studies, without any comparison between the studies; surgical philosophy is not discussed at all (decompression surgery, decompression + stabilization), and one very important clinical aspect - not only time of the surgery in terms of from onset of symptoms to surgery, but also time of the day of the surgery - were there any studies which analyze this very important aspect and its correlation to timing? - for example:

Baig Mirza A, Velicu MA, Lyon R, Vastani A, Boardman T, Al Banna Q, Murphy C, Kellett C, Vasan AK, Grahovac G. Is Cauda Equina Surgery Safe Out-of-Hours? A Single United Kingdom Institute Experience. World Neurosurg. 2022 Mar;159:e208-e220. doi: 10.1016/j.wneu.2021.12.028. Epub 2021 Dec 14. PMID: 34915208.

Demetriades AK. Cauda equina syndrome - from timely treatment to the timing of out-of-hours surgery. Acta Neurochir (Wien). 2022 May;164(5):1201-1202. doi: 10.1007/s00701-022-05174-1. Epub 2022 Mar 30. PMID: 35352153.

Are there any information of complication rates of these surgeries? To simply describe timing without description of outcome does not make a lot of sense, so I suggest to include outcome assesment following surgery in correlation to time, and to include into Discussion and comment on current literature:

Woodfield J, Lammy S, Jamjoom AA, Fadelalla MA, Copley PC, Arora M, Glasmacher SA, Abdelsadg M, Scicluna G, Poon MT, Pronin S, Leung AH, Darwish S, Demetriades AK, Brown J, Eames N, Statham PF, Hoeritzauer I; UCES Study Collaborators; British Neurosurgical Trainee Research Collaborative. Demographics of Cauda Equina Syndrome: A Population Based Incidence Study. Neuroepidemiology. 2022 Oct 31. doi: 10.1159/000527727. Epub ahead of print. PMID: 36315989.

I urge authors to analyze the following manuscript and compare their results to this review:

Epstein NE. Review/Perspective: Operations for Cauda Equina Syndromes - "The Sooner the Better". Surg Neurol Int. 2022 Mar 25;13:100. doi: 10.25259/SNI_170_2022. PMID: 35399881; PMCID: PMC8986648.

A: Thank you for reviewing our manuscript and providing the above comments. Following the updated search, we have now included the above three articles in our review. The scope of this study was to address the definition of CES and how timing of CES is reported. Aetiology and surgical procedures undertaken were not discussed considering volume of content reported and readability. Thank you for highlighting this as it is an area, we can focus our future work on. We have now added to the role of time of surgery in the discussion. We hope this provides readers with more information about CES and adds to the evidence regarding the heterogeneity of the condition and the current lack of consensus with its definition. This should further emphasise the need for a consistent definition of CES and agreement for the time-point at which time to surgery should be measured from.

---

## [Decision Letter · Decision Letter 2]

20 Feb 2023

PONE-D-22-00142R2Definition and surgical timing in Cauda Equina Syndrome – An updated systematic reviewPLOS ONE

Dear Dr. Gillespie,

Thank you for submitting your manuscript to PLOS ONE. After careful consideration, we feel that it has merit but does not fully meet PLOS ONE’s publication criteria as it currently stands. Therefore, we invite you yet again to submit a revised version of the manuscript that addresses the points raised during the review process. One reviewer has made specific recommendations which we hope can be addressed.

We look forward to receiving your revised manuscript.

Kind regards,

Andreas K Demetriades, MBBChir, MPhil, FRCSEd, FEBNS.

Academic Editor

PLOS ONE

Reviewers' comments:

Reviewer's Responses to Questions

**Comments to the Author**

1. If the authors have adequately addressed your comments raised in a previous round of review and you feel that this manuscript is now acceptable for publication, you may indicate that here to bypass the “Comments to the Author” section, enter your conflict of interest statement in the “Confidential to Editor” section, and submit your "Accept" recommendation.

Reviewer #3: All comments have been addressed

Reviewer #4: All comments have been addressed

2. Is the manuscript technically sound, and do the data support the conclusions?

Reviewer #3: Yes

Reviewer #4: Yes

3. Has the statistical analysis been performed appropriately and rigorously? 

Reviewer #3: Yes

Reviewer #4: N/A

4. Have the authors made all data underlying the findings in their manuscript fully available?

Reviewer #3: Yes

Reviewer #4: Yes

5. Is the manuscript presented in an intelligible fashion and written in standard English?

Reviewer #3: Yes

Reviewer #4: Yes

6. Review Comments to the Author

Reviewer #3: (No Response)

Reviewer #4: Authors have provided a revised version of their manuscript on timing of surgery in cases of cauda equine syndrome. Recommended literature has been included and discussed. Authors state that due to the volume of the analyzed studies, further inquiry regarding etiology which led to cauda syndrome as well as the role of surgical therapy could not have been analyzed. I urge authors to go through the studies and look at this issue, even more data of descriptive nature could be of interest and enrich the manuscript. Especially timing in relation to trauma and tumors has a different significance and dynamics.

7. PLOS authors have the option to publish the peer review history of their article (what does this mean?). If published, this will include your full peer review and any attached files.

Reviewer #3: No

Reviewer #4: No

---

## [Author Response · Author response to Decision Letter 2]

4 Apr 2023

Conor S. Gillespie

Institute of Systems, Molecular and Integrative Biology

University of Liverpool

Liverpool 

L69 7BE, UK 

2nd April 2023

Dr. Andreas K. Demetriades, 

Academic Editor, 

PLOS One 

Dear Dr. Demetriades,

Re: Definition and surgical timing in Cauda Equina Syndrome – An updated systematic review 

Thank you considering our revised manuscript entitled “Definition and surgical timing in Cauda Equina Syndrome – A systematic review” for publication in PLOS One. We thank the reviewers for their constructive comments and feedback. We hope we have addressed their concerns adequately in our revision of the manuscript and attach our point-by-point responses below. We believe that this manuscript is an important addition to the literature and the revisions have strongly enhanced this message. 

I would like to confirm again that with this submission, all study authors declare that they have reviewed the final manuscript, approved the contents of it and that the requirements for authorship have been met by all named authors. We can guarantee that the work is not being considered for publication by any other journal and that no previous work we’ve published (apart from meeting abstracts) overlap with the current work. We have not received any external funding for the completion of this work and have no relevant conflicts of interest to report.

I thank you once again for considering our manuscript for publication and look forward to hearing from you soon. 

Yours Sincerely,

Conor S. Gillespie 

Honorary Clinical Associate 

Institute of Systems, Molecular and Integrative Biology (ISMIB) 

University of Liverpool, UK 

Email: hlcgill2@liv.ac.uk

Response to review comments to the author:

Reviewer #4: Authors have provided a revised version of their manuscript on timing of surgery in cases of cauda equine syndrome. Recommended literature has been included and discussed. Authors state that due to the volume of the analyzed studies, further inquiry regarding etiology which led to cauda syndrome as well as the role of surgical therapy could not have been analyzed. I urge authors to go through the studies and look at this issue, even more data of descriptive nature could be of interest and enrich the manuscript. Especially timing in relation to trauma and tumors has a different significance and dynamics.

A: Thank you for raising the above point regarding addition of the CES etiology and role of surgical therapy to our manuscript. We now have a descriptive section outlining the underlying etiology of CES (Results, Line 171-175). We have also included a section in our results, titled “Surgical Method”, Line 217-229, describing the predominant surgical method employed for studies included. In addition, our discussion (Line 275-279) now acknowledges the role of CES etiology in the heterogeneity of this condition. We hope these changes have strengthened our review and provided readers with relevant information regarding the CES etiology and surgical methods reported in the literature.

---

## [Decision Letter · Decision Letter 3]

14 Apr 2023

Definition and surgical timing in Cauda Equina Syndrome – An updated systematic review

PONE-D-22-00142R3

Dear Mr Gillespie,

We’re pleased to inform you that your manuscript has been judged scientifically suitable for publication and will be formally accepted for publication once it meets all outstanding technical requirements.

Kind regards,

Andreas K Demetriades, MBBChir, MPhil, FRCSEd, FEBNS.

Academic Editor

PLOS ONE

Additional Editor Comments (optional):

Congratulations on addressing all peer review points during this lengthy process.

Reviewers' comments:

Reviewer's Responses to Questions

**Comments to the Author**

1. If the authors have adequately addressed your comments raised in a previous round of review and you feel that this manuscript is now acceptable for publication, you may indicate that here to bypass the “Comments to the Author” section, enter your conflict of interest statement in the “Confidential to Editor” section, and submit your "Accept" recommendation.

Reviewer #3: All comments have been addressed

Reviewer #4: All comments have been addressed

2. Is the manuscript technically sound, and do the data support the conclusions?

Reviewer #3: Yes

Reviewer #4: Yes

3. Has the statistical analysis been performed appropriately and rigorously? 

Reviewer #3: Yes

Reviewer #4: Yes

4. Have the authors made all data underlying the findings in their manuscript fully available?

Reviewer #3: No

Reviewer #4: Yes

5. Is the manuscript presented in an intelligible fashion and written in standard English?

Reviewer #3: Yes

Reviewer #4: Yes

6. Review Comments to the Author

Reviewer #3: (No Response)

Reviewer #4: Authors provide a revised version of their literature review on definition and surgical timing in cauda equina surgery. Although descriptive, section on underlying etiology has been added to the review. Authors have sufficiently responded to reviewer remarks.

7. PLOS authors have the option to publish the peer review history of their article (what does this mean?). If published, this will include your full peer review and any attached files.

Reviewer #3: No

Reviewer #4: No

---

## [Editor Report · Acceptance letter]

24 Apr 2023

PONE-D-22-00142R3 

Definition and surgical timing in Cauda Equina Syndrome – An updated systematic review 

Dear Dr. Gillespie:

I'm pleased to inform you that your manuscript has been deemed suitable for publication in PLOS ONE. Congratulations! Your manuscript is now with our production department. 

Kind regards, 

on behalf of

Dr. Andreas K Demetriades 

Academic Editor

PLOS ONE